# Thermal Analysis of Parylene Thin Films for Barrier Layer Applications

**DOI:** 10.3390/polym14173677

**Published:** 2022-09-04

**Authors:** Sébastien Buchwalder, Aurelio Borzì, Juan J. Diaz Leon, Florian Bourgeois, Cléo Nicolier, Sylvain Nicolay, Antonia Neels, Olaf Zywitzki, Andreas Hogg, Jürgen Burger

**Affiliations:** 1Sitem Center for Translational Medicine and Biomedical Entrepreneurship, University of Bern, Freiburgstrasse 3, 3010 Bern, Switzerland; 2Graduate School for Cellular and Biomedical Sciences, University of Bern, Mittelstrasse 43, 3012 Bern, Switzerland; 3Center for X-ray Analytics, Swiss Federal Laboratories for Materials Science and Technology, Empa, Überlandstrasse 129, 8600 Dübendorf, Switzerland; 4CSEM Sustainable Energy Center, Jaquet-Droz 1, 2002 Neuchâtel, Switzerland; 5Coat-X SA, Eplatures-Grise 17, 2300 La Chaux-de-Fonds, Switzerland; 6Fraunhofer Institute for Organic Electronics, Electron Beam and Plasma Technology (FEP), Winterbergstrasse 28, 01277 Dresden, Germany

**Keywords:** parylene, vapor phase deposition, annealing, thermal stability, water vapor transmission rate (WVTR), helium transmission rate (HTR)

## Abstract

Biocompatible polymer films demonstrating excellent thermal stability are highly desirable for high-temperature (>250 °C) applications, especially in the bioelectronic encapsulation domain. Parylene, as an organic thin film, is a well-established polymer material exhibiting excellent barrier properties and is often the material of choice for biomedical applications. This work investigated the thermal impact on the bulk properties of four types of parylene films: parylene N, C, VT4, and AF4. The films, deposited using the standard Gorham process, were analyzed at varying annealing temperatures from room temperature up to 450 °C. Thermal properties were identified by differential scanning calorimetry (DSC) and thermogravimetric analysis (TGA) methods, while X-ray diffraction (XRD) analysis showed the effect of high-temperature exposure on the structural properties. In addition to thermal and structural analysis, the barrier properties were measured through the helium transmission rate (HTR) and the water vapor transmission rate (WVTR). Fluorinated parylene films were confirmed to be exceptional materials for high-temperature applications. Parylene AF4 film, 25um thick, demonstrated excellent barrier performance after 300 °C exposure, with an HTR and a WVTR of 12.18 × 10^3^ cm^3^ (STP) m^−2^ day^−1^ atm^−1^ and 6.6 g m^−2^ day^−1^, respectively.

## 1. Introduction

Polymeric materials are often used for packaging or encapsulation solutions; however, common solutions using bulk packaging express limitations in terms of miniaturization and barrier performance [1]. That is why thin films, exhibiting excellent barrier properties, are used in many polymer applications today. Poly-para-xylylene-based polymers, also known as parylenes, are polymers that can be deposited in thin film form at room temperature by chemical vapor deposition. The deposition process allows pinhole-free growth without inducing internal stress and possesses a high step coverage [2]. Due to the unique properties of the film, parylene is considered the material of choice for various domains, such as bio-MEMS [3,4,5,6,7], electronic insulation [8,9], energy harvesting [10,11,12], flow and temperature sensors [13,14,15] or medical device encapsulation [2,16,17,18]. Furthermore, a parylene film demonstrating excellent thermal stability is highly desirable for high-temperature environment applications, such as aerospace, automotive, battery, or high-power electronic fields. Moreover, a thin film able to resist the reflow soldering process, during which the maximal temperature can reach 270 °C [19,20], could provide an excellent protection solution for electronic circuits and components. Higher thermal stability demonstrated by parylene films would as well confirm parylene as a potential substrate material for printed circuit boards (PCBs). Finally, parylene films, often cited as the gold standard material for medical applications due to their high biocompatibility and biostability, demonstrating superior temperature stability, are required to resist multiple steam sterilization cycles up to 135 °C [21,22] for implantable applications.

In this study, four parylene types (N, C, VT4, AF4) were selected and characterized. The chemical structure of each type is illustrated in Figure 1.

The aim of this study was to investigate the effects of two different postdeposition heat treatments on the structural, physical, and functional properties of the above four parylene types. Thermal stability and transitions of interest were evaluated by differential scanning calorimetry (DSC) analysis and thermogravimetric analysis (TGA). X-ray diffraction (XRD) analysis was carried out to determine d-spacing, defined as interatomic spacing, and to quantify the size of the crystalline phase. In addition, permeation measurements were conducted to determine the helium transmission rate (HTR) and water vapor transmission rate (WVTR). The permeation measurements allowed to quantify the barrier properties depending on the heat treatments. In the second Section, the parylene types and deposition process are presented, followed by the different heat treatments applied and a description of the characterization methods employed in this study. The results are presented in Section 3 and discussed in the following Section. Finally, the conclusion summarizes the results and expresses some perspectives.

## 2. Materials and Methods

### 2.1. Parylene Film Deposition

The parylene films were deposited at room temperature by low-pressure chemical vapor deposition (LPCVD) based on the Gorham process [23]; the depositions were performed at the company Coat-X SA (Switzerland). Four parylene types were deposited: Parylene N, poly(*p*-xylylene), is the basic form of parylene, consisting of a linear carbon–hydrogen molecule structure. Showing a low dielectric constant and assuming a high degree of crystallinity, parylene N is especially suitable for high-frequency electronic applications [24,25]. Parylene C, poly(chloro-*p*-xylylene), is the most widespread parylene form. Parylene C offers excellent barrier protection due to its low permeability regarding gas and moisture [26], and it is commonly established to permit a high deposition rate [3]. Finally, parylene VT4 and AF4, poly(tetrafluoro-*p*-xylylene), also called parylene F, are the most recently synthesized parylene dimers. Parylene VT4 incorporates fluorine atoms in the aromatic sites, while parylene AF4 replaces the α hydrogen atoms with fluorine at the end of its aromatic ring. Both parylene VT4 and AF4 are known to possess superior thermal stability and a low dielectric constant [27]. In addition, fluorinated polymers and, more specifically, parylene AF4 propose an increased thermal and oxidative stability and a high resistance to UV exposure [28,29,30]. Lastly, parylene AF4 also exhibits the highest penetrating ability of the parylene types and the lowest friction coefficient [31,32]. 

P-type silicon wafers (100), previously rinsed with acetone and isopropanol solutions in an ultrasonic bath for 1 min each and dried with high purity nitrogen, were used as a substrate for the XRD analysis, while self-standing parylene membranes were prepared for the other characterization methods. On the silicon wafers samples, an oxygen plasma was applied for 30 s with 50 W power at 13.5 MHz frequency in order to remove traces of residual hydrocarbons on the surface and to improve the adhesion of parylene films by the creation of attachment sites [27]. The parylene deposition process itself involved three steps. In the first step, the solid dimer was vaporized in the gas phase using temperatures comprised in the range of 80–120 °C at a pressure of about 0.1 mbar. The second step, called pyrolysis, cleaved the dimers in monomers at temperatures between 650 and 770 °C. The sublimation and pyrolysis temperatures differ slightly for the used parylene forms. Finally, the monomers condensed and polymerized in the chamber to form a polymeric film. A cold trap was placed before the pumping system to capture the excess parylene and to protect the vacuum pump system.

### 2.2. Heat Treatments

After parylene deposition, the free-standing membranes and silicon wafer samples were analyzed directly or aged with two different postdeposition heat treatments. First, an annealing process, long-term high-temperature exposure under a nitrogen environment, was performed using an RTP-1000-150 furnace from Unitemp GmbH, Pfaffenhofen/Ilm, Germany. Different annealing temperatures were completed to determine the effects of the temperature on parylene films. The film characterization started at 25 °C, and then the films were annealed from 50 °C up to 450 °C by steps of 50 °C. The duration of the treatment was 1 h 15 min at the set temperature.

The second treatment was a reflow soldering process performed under an oxygen-containing environment. An annealing treatment under an inert environment reduces the chemical modifications of the films, but a reflow soldering process under an atmospheric environment is more representative of final applications. The temperature profile was programmed and executed by ProtoFlow S N2 convection oven from LPKF Laser & Electronics AG, Garbsen, Germany, made for lead-free reflow soldering of rapid PCB soldering applications. In order to match the different standards (e.g., IPC/JEDEC J-STD-020) that describe various soldering processes, a duration of 5 min at the maximum temperature was applied. In this case, the maximal process temperature was determined for each parylene type.

### 2.3. Differential Scanning Calorimetry

Differential scanning calorimetry (DSC) method, measuring the heat flux variations due to thermodynamic transformations, was used to identify the phase transitions. The DSC equipment (DSC 214 Polyma from NETZSCH-Gerätebau GmbH, Selb, Germany) applied a temperature range from 40 to 470 °C using a heating and a cooling ramp of 10 K/min under a nitrogen environment. The amount of film material was about 5 mg.

### 2.4. Thermogravimetric Analysis

Thermogravimetric analysis (TGA), which measures weight changes in a material as a function of temperature under a controlled atmosphere, was used to determine the decomposition and transformation temperatures of our parylene films. For that purpose, the films were heated up to 800 °C with a heating ramp of 10 K/min under a nitrogen environment. The equipment, TG 209 F1 Libra (NETZSCH-Gerätebau GmbH, Selb, Germany), used for the measurements requires a minimum amount of film material of 5 mg.

### 2.5. X-ray Diffraction

X-ray diffraction measurements were performed at room temperature with a Malvern PanAlytical Empyrean instrument used in Bragg-Brentano geometry using the Cu Kα = 1.5418 Å radiation and optics at the incident and diffracted beam paths. Asymmetric noncoplanar scans of 2 Theta were performed in grazing incidence; the incidence angle was fixed to 0.5° to maximize the signal from the thin film. Information such as crystalline or amorphous status, size of crystalline domains, and strain was extracted from the analysis. Data treatment and analysis were performed by using HighScore Plus software [33].

### 2.6. Helium Gas Permeation

The helium gas permeation was quantified using an experimental setup developed by Hogg et al. [34]. The principle of the measurement is based on quadrupole mass spectrometry detection. Self-standing parylene membranes were introduced in a vacuum system with adjustable gas pressure on one side of the membrane. A defined supply pressure of helium gas was maintained in the test volume using an injection valve and a bypass pump. Before the measurements, the setup was calibrated thanks to a calibrated leak made of sintered stainless steel, previously presented by Yoshida et al. [35,36]. This element, exhibiting a constant conductance due to its porous structure (pore sizes of less than 1µm), was used to generate a reference molar flux for in situ calibration of the quadrupole mass spectrometer detection. The gas that passed through the constant conductance element (CCE) satisfied the molecular flow conditions from pressures between 10 mbar and 100 mbar. Once the molecular conductance of the element was calibrated with nitrogen gas, the molar flux could be calculated for different gases, including helium, as reported in [37]. Equation (1) describes how the helium flow rate (Q_He_) in mol/s was obtained:(1)$$Q_{\mathrm{He}}{=C}_{N_{2}}\;\sqrt{\frac{M_{N_{2}}}{M_{\mathrm{He}}}}\;\frac{P_{R}}{R\;\sqrt{{T\;T}_{C}}}\;,\;$$
where ${\;C}_{N_{2}}$(m^3^/s), given by the supplier, is the molecular conductance for N_2_ of the CCE, and $M_{N_{2}}$ and $M_{\mathrm{He}}$ are the molar mass of N_2_ (28 g/mol) and He (4 g/mol), respectively. P_R_ (in Pa) is the upstream pressure in the gas reservoir, and R (J mol/K) is the gas constant. T is the temperature of the gas applied during the measurement, and T_C_ is the temperature used during the calibration of $C_{N_{2}}$, performed by the supplier. Thanks to this calibration, the relationship between helium molar flux and the mass spectrometer measure, the ion current (A), could then be established, and, hence, the amount of helium gas molecules passing through the test membranes could be quantified. The helium transmission rate (HTR) (cm^3^ (STP) m^−2^ day^−1^ atm^−1^) was then calculated by: (2)$$\mathrm{HTR}=\frac{Q_{\mathrm{He}}{\;R\;T}_{0}{\;10}^{6}\;24\;3600}{P_{\mathrm{atm}}\;A\;\Delta P\;},\;$$
where Q_He_ (mol/s) is the helium flow rate, calculated above, R (J mol/K) is the gas constant, and T_0_ (K) and P_atm_ are the temperature and pressure at the standard condition, namely 273.15 K and 10^5^ Pa. A (m^2^) is the effective surface area of the membrane, and ΔP (in atm) is the different pressure between the two sides of the membrane; ΔP can be approximated by the helium supply pressure applied on one side of the membrane as suggested by Yoshida et al. [38]. The supply pressure was measured by a capacitive pressure gauge and fixed to 18 mbar at 25 °C during the measurements to avoid inelastic membrane deformation.

### 2.7. Water Vapor Permeation

The water vapor transmission rate was determined by the electrolytic detection sensor method with instruments of type WDDG (Brugger Feinmechanik GmbH)) according to the international standard ISO 15106-03 and performed in collaboration with the Fraunhofer Institute for Organic Electronics, Electron Beam and Plasma Technology (FEP), located in Dresden. The sample, with a diameter of 100 mm (measurement area 78.5 cm^2^), was inserted into a test cell with two chambers: a dry chamber and a controlled humidity chamber. In the controlled humidity chamber, a sulfuric acid solution delivered a constant water vapor pressure. The coated side of the sample was directed to the dry chamber with the sensor. The water vapor permeating through the specimen was carried by the dry nitrogen carrier gas into the electrolytic cell. This cell contains two spiral wire electrodes coated with a thin layer of phosphorous pentoxide. The water vapor in the carrier gas was electrolytically decomposed into hydrogen and oxygen by the application of a DC voltage. The mass of the permeating moisture per time interval was calculated from the electrolytic current and per area of the test specimen. The water vapor transmission rate was calculated by the following equation:(3)$$\mathrm{WVTR}=\frac{\;I\;}{A}\;8.067\;,$$
where WVTR (g m^−2^ day^−1^) is the water vapor transmission rate of the specimen, expressed in grams per square meter and per day, A (m^2^) is the transmission area of the test specimen in square meters, I (A) is the electrolytic current in amperes, and 8.076 is the instrument constant. The water vapor transmission rate was measured at a temperature of 38 °C and relative humidity of 90 %. The usual WVTR measurement range is between 0.001 and 10 g m^−2^ day^−1^; if necessary, higher values can be determined by reducing the measurement area.

## 3. Results

### 3.1. Differential Scanning Calorimetry

Differential scanning calorimetry analysis permitted to identify the phase transitions of all four parylene types, as illustrated in Figure 2.

Three phase transitions were found for parylene N. Figure 2a shows a sudden heat flow change in the first heating curve at 430 °C, which corresponds to the melting point (endothermic process). The melting temperature has been reported previously between 410 and 430 °C [25,39,40,41]. In addition to the melting point, two exothermic phase transitions were determined for parylene N in the first heating curve: the first at 228 °C, which corresponds to the transition from the monoclinic α phase to a hexagonal β1 phase, and the second occurs at 281 °C, from hexagonal β1 to β2, as formerly observed and described in detail by Miller et al. [39]. The difference between β1 and β2 phases is still not completely understood, but previous works suggest a shortening of the *c* axis of the unit cell [39,40]. Parylene VT4, illustrated in Figure 2b, exhibits a melting point at 435 °C, but the polymer starts to degrade above 300 °C. Parylene VT4 thermal properties are difficult to be confirmed since parylene VT4 has been synthesized recently compared with the other types, and related studies are rare. Nevertheless, Senkevich et al. found a melting point at 402 °C [42]. The melting point of parylene C has been detected at 301 °C, as illustrated in Figure 2c; that value is close to 290 °C, as mentioned in the literature [42,43]. Finally, in Figure 2d, no phase transition was identified for parylene AF4 after one heat cycle up to 470 °C. However, during the cooling process, the polymer reveals a crystallization point at 395 °C, and the second heating cycle indicates a melting point at 406 °C. That result can be related to an observed reversible crystalline phase transition around 400 °C [40]. A second heat cycle (orange curves) was applied to each parylene type, but it did not reveal additional information, except for parylene AF4, as mentioned above. 

### 3.2. Thermogravimetric Analysis

Thermogravimetric analysis allowed to determine the thermal decomposition points for parylene films. Figure 3 illustrates the residual mass, in percentage, as a function of temperature.

Nonfluorinated films parylene N and parylene C show a decomposition temperature at 489 °C and 503 °C, respectively. The degradation of parylene VT4 occurs at 495 °C. In a previous study, Kahouli et al. established the thermal degradation of parylene VT4 at 515 °C [27]. As the degradation temperatures do not greatly vary between nonfluorinated parylene N and C and fluorinated parylene VT4, we can exclude the role of fluorine bonds in this degradation process. In fact, we can hypothesize that the degradation originates from the cleavage of C–H aliphatic stretching. In contrast, our parylene AF4 layer shows the highest thermal stability, with a decomposition temperature established at 568 °C, even if a slight loss of weight can be observed after 330 °C. Our fabricated parylene AF4 layer indicates a higher decomposition temperature compared to the literature, in which it has been estimated to be slightly above 500 °C [40,44].

### 3.3. X-ray Diffraction

The X-ray diffraction analysis allows to extract parameters of the parylene structural properties at the atomic scale. The d-spacing (Å), obtained from the Bragg equation, represents, in this particular case, the distance between two benzene rings of the parylene monomer in a crystalline region. In addition, the Scherrer equation can be used to obtain the size of the crystalline domain in the film:(4)$$D=\frac{k\;\lambda \;}{\beta \;\cos(\theta )},$$
where D is the size of the crystalline domain (Å), k is the shape factor (0.9), β is the full width at half maximum of the intensity peak (FWHM), λ (nm) is the X-ray wavelength, and θ (rad) is the diffraction angle. Those two parameters are illustrated in Figure 4 for each parylene type.

Figure 4a compares the value of the d-spacing as a function of the annealed temperature for each parylene type. We observe that only parylene N shows a sudden change after 250 °C, whereas the values for other parylene films do not indicate sudden changes. This major structural change of parylene N after 250 °C can be explained by the transition between monoclinic α to hexagonal β phase revealed by DSC analysis. The evolution of the size of the crystalline domains, illustrated in Figure 4b, reveals that parylene N and C undergo a process of increasing the structural order as a function of the annealed temperature, while parylene F, VT4, and AF4, do not show any remarkable changes. Ordered domains extend over the largest distances in parylene N when as-deposited polymers are considered. Parylene C has the most noticeable ordering process upon heat exposure. It should finally be noted that parylene C films were characterized up to a maximum of 250 °C due to their heat resistance limitation. Indeed, parylene C proved minor heat resistance, with XRD disappearing after annealing at 300 °C. Finally, the XRD results, combined with the DSC analysis, lead to consider fluorinated parylene and, more especially, the AF4 type as mainly amorphous.

Finally, nonfluorinated parylene films show the largest crystalline domains, with 179.5 Å and 123.3 Å after the annealing treatment, respectively, for parylene N and C. They also exhibit the highest increase in the crystalline domain size from the as-deposited and after heat treatment comparison.

### 3.4. Permeation Measurements

Two different permeation measurement techniques were used to quantify the parylene films’ permeability: the helium transmission rate (HTR) and the water vapor transmission rate (WVTR). Those two values were evaluated for three different conditions: as-deposited (no post-treatment applied), after the annealing process, and after the reflow soldering process under an oxygen-containing environment. The black columns show the results when no post-treatment was applied, while the blue and the red columns display the results after the annealing and reflow processes were performed, respectively. The annealing temperature is 300 °C for all parylene types (except 250 °C for parylene C) under a nitrogen environment. The reflow temperatures are 150 °C, 200 °C, and 300 °C for parylene N, C, and both F, respectively, under an oxygen-containing environment. The results presented in Figure 5 are normalized to a 25 µm thick parylene film, and the applied treatment temperature for each parylene film is written below the column.

As illustrated in Figure 5, all parylene types, except for parylene VT4, show an improvement in barrier performances regarding helium gas and water vapor transmission rate after heat treatments. Parylene N shows the largest improvement factor after exposure at 300 °C under a nitrogen environment for both transmission rates, but its use is limited to applications below 150 °C under an oxygen-containing environment. Parylene C exhibits the best barrier properties of all the parylene types, with an HTR of 2.21 × 10^3^ cm^3^ (STP) m^−2^ day^−1^ atm^−1^ and a WVTR 2.8 g m^−2^ day^−1^ if no heat treatment is applied. However, the maximal use temperature of parylene C is restrained to 200 °C under an oxygen-containing environment to avoid the degradation of the film. Two different behaviors occur for fluorinated parylene films. Whereas parylene AF4 describes the same tendency as nonfluorinated parylene films, parylene VT4 sightly loses its barrier performance after high-temperature exposure. Furthermore, parylene AF4 indicates the best barrier properties after heat exposure at 300 °C regarding helium gas or water vapor hermiticity, demonstrating an HTR of 12.18 × 10^3^ cm^3^ (STP) m^−2^ day^−1^ atm^−1^ and a WVTR of 6.6 g m^−2^ day^−1^. Finally, the heat-treated parylene AF4 WVTR values at 300 °C under an oxygen-containing environment are similar to parylene C values, limited to 200 °C.

## 4. Discussion

The polymer film structures show both crystalline or ordered domains and amorphous regions. It means that parylene film can be defined as a partially crystalline or semicrystalline material [41]. As demonstrated in former studies, annealing treatment for parylene allows the polymer chains to form crystalline domains in an amorphous region [45]; these former observations are in agreement with our study, especially for nonfluorinated parylene types. Our structural analyses confirm the temperature effect for parylene N and C, resulting in a significant increase in the crystalline size domain of 130% and 147%, respectively, for parylene N and C films, whereas the size of the crystalline domain remains constant over the temperature range for both parylene F. As shown in Figure 5, the increase in film crystallinity degree has a beneficial impact on the barrier performances of the films. Indeed, an annealing treatment lower than the melting point improves the barrier performances regarding the HTR and the WVTR for all parylene types, except parylene VT4. Furthermore, it is believed that this increase also affects the film properties, such as mechanical and electrical properties [42]; however, a superior crystallinity degree can introduce tensile thermal stress due to polymer shrinkage [46]. In addition, microstructural displacements are not suitable when parylene films are combined with inorganic layers to build a multilayer stack. Surface morphology variations and/or internal stress formation induced by the annealing process can lead to a loss of barrier performance of the parylene-based multilayer. Thanks to the permeation measurements, the relationship between thermal (DSC and TGA), structural (XRD), and functional (HTR and WVTR) properties of parylene films could be understood to a certain level. The HTR and the WVTR characterization methods allowed to quantify the permeation properties of parylene film as a function of the annealing temperature. Parylene types N and C are limited in heat treatment under an oxygen-containing environment to 150 °C and 200 °C, respectively. Above those values, the films become brittle and slightly yellowish due to oxidation. On the contrary, parylene VT4 and AF4 do not show any degradation up to 300 °C. Moreover, parylene AF4 demonstrates a decomposition temperature of 568 °C, about 70 °C higher than mentioned in the literature [40,44]. This difference can be explained by a lower pyrolysis temperature (650 °C) applied during the deposition process, and, hence, it can lead to incomplete cleaving of the dimer in monomer. The thermal- and atomic-scale structural stabilities of fluorinated parylenes confirm their great potential for high-temperature applications. Thanks to its high thermal stability, the parylene AF4 type displays similar WVTR values at 300 °C to parylene C at 200 °C under an oxygen-containing environment.

In comparison to common polymer materials, fluorinated parylene films exhibit a very high thermal stability in addition to low permeability. Most polymers available on the market have limitations in terms of heat resistance and/or barrier performance. Epoxy, polyurethane, or acrylic solutions present much lower barrier performance and lower heat resistance [1]. Polyethylene terephthalate (PET) Melinex 401 CW type and polyethylene naphthalate (PEN) Teonex HV, both produced by DuPont Teijin Films, propose a melting point respectively between 255 °C and 260 °C for PET and at 263 °C for PEN, as specified by the product datasheets. WVTR values for a 25 µm thick film of PET Melinex and PEN Teonex are 23.7 and 5.7 g m^−2^ day^−1^, respectively. Polyimide (PI), DuPont Kapton, one of the most thermally stable polymers with a decomposition point around 500 °C [47], presents a helium transmission rate of 2.3 × 10^5^ cm^3^ m^−2^ day^−1^ atm^−1^ at 245 °C [48] for a thickness film of 25 µm, while both parylene VT4 and AF4 yield values 10 times smaller at 2.9 × 10^4^ cm^3^ m^−2^ day^−1^ atm^−1^ and 1.7 × 10^4^ cm^3^ m^−2^ day^−1^ atm^−1^ at 300°C, respectively. Moreover, parylene AF4, even after exposure at 300 °C, outclasses PI in terms of water vapor barrier performance, with a WVTR value of 6.6 g m^−2^ day^−1^ against 54 g m^−2^ day^−1^ for PI film [48] under the same test conditions (38 °C and 90% R.H.).

## 5. Conclusions

The results presented in this study demonstrate the impact of thermal treatment on the physical, structural, and functional properties of different parylene types. DSC and TGA analysis showed that phase transition identification and decomposition temperatures provide essential information to interpret XRD data. X-ray diffraction investigations supplied information regarding the annealing effect at the atomic-scale structure. The results show that nonfluorinated parylene crystallinity greatly increases with annealing temperature, while the degree of crystallinity of fluoridated parylene films remains constant. Both permeation measurements prove that an increase in crystallinity improves the barrier properties, quantified by the helium gas and water vapor transmission rate. However, an increase in crystallinity might induce microstructural displacements in the film and can lead to undesirable changes such as surface morphology variations and/or internal stress build-up. Permeation measurements, performed in two different aging conditions, confirm that parylene C performs best if the temperature does not exceed 200 °C in an oxygen-containing environment. For higher temperature applications, parylene VT4 and AF4 reveal the best barrier performances. Finally, parylene AF4 demonstrates very low permeability against helium gas and water vapor, even after high-temperature exposure in an environment containing oxygen, even better than common polymeric materials available on the market. Furthermore, 300 °C reflow processed parylene AF4 film exhibits WVTR values close to 200 °C treated parylene C films. These results validate that fluorinated parylene films could be an appropriate material for high-temperature environments, such as aerospace, automotive, battery, or high-power electronic fields. Moreover, as parylene VT4 and AF4 are able to resist the reflow soldering process until 300 °C, these parylene types could provide an excellent protection solution for electronic circuits and components. Higher thermal stability expressed by fluorinated parylene films also confirms parylene as a potential substrate material for PCBs. More specifically, parylene AF4 represents an exceptional material in the high-temperature encapsulation domain. In addition to its ultrahigh thermal stability, parylene AF4 was demonstrated to be completely biostable and biocompatible [49]. These properties together could make it somewhat the gold standard for biomedical encapsulation solutions, enabling a new field of parylene thin film applications that require higher heat resistance.

## Figures and Tables

**Figure 1 polymers-14-03677-f001:**
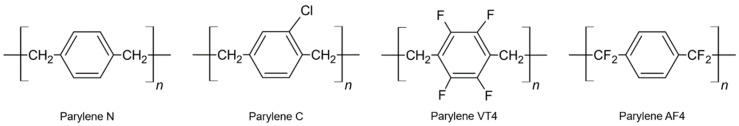
Chemical structure of parylene type N, C, VT4, and AF4.

**Figure 2 polymers-14-03677-f002:**
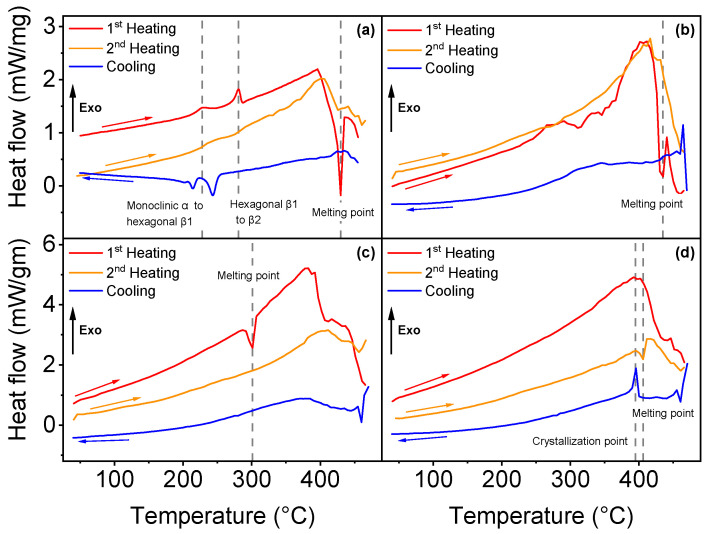
DSC analysis curves for each parylene type: (**a**–**d**) illustrate the first and second heating cycles and the cooling curves, respectively, for parylene N, VT4, C and AF4.

**Figure 3 polymers-14-03677-f003:**
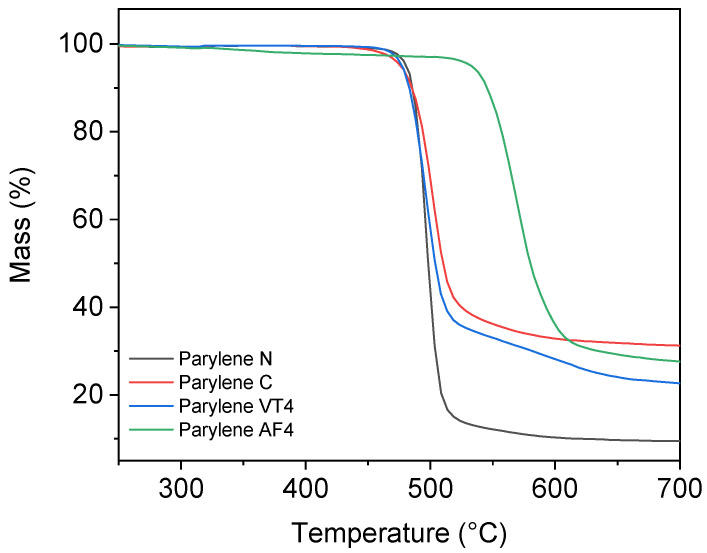
Thermogravimetric analysis results of parylene type N, C, VT4, and AF4.

**Figure 4 polymers-14-03677-f004:**
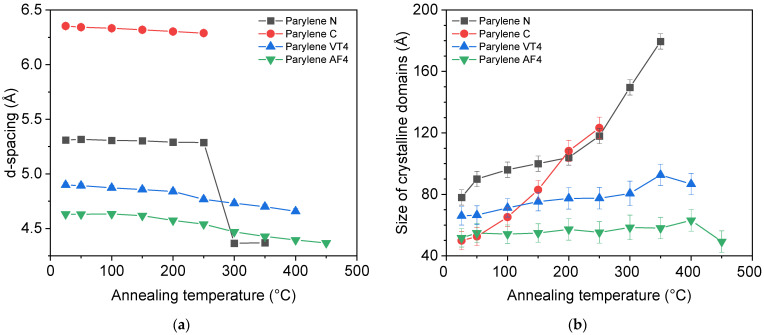
Atomic-scale structural analysis performed at room temperature for each type of parylene: (**a**) the d-spacing (Å) as a function of the annealed temperatures; (**b**) the size of the crystalline domains (Å) as a function of the annealed temperatures.

**Figure 5 polymers-14-03677-f005:**
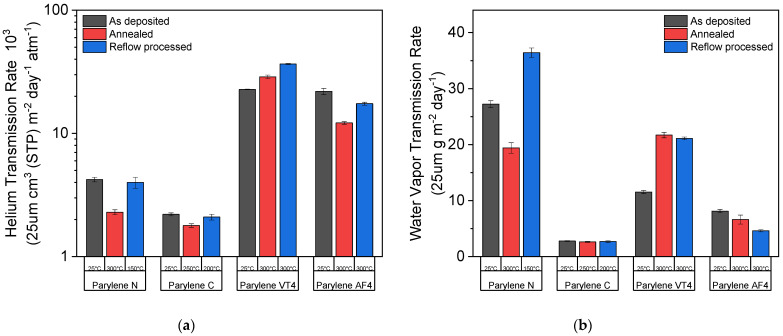
Permeation measurements: (**a**) helium transmission rate (HTR) and (**b**) water vapor transmission rate (WVTR) as a function of temperatures and for the different parylene types.

## Data Availability

The data presented in this study are available on request from the corresponding author.

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
