# Peer review of "Thermal Analysis of Parylene Thin Films for Barrier Layer Applications"

_polymers, 2022, doi:10.3390/polym14173677_

Round 1

Reviewer 1 Report

The paper deals with the investigates the thermal impact on bulk properties for four types of parylene films. Parylene, as an organic thin film, is a well-established polymer material exhibiting excellent barrier properties and being often the material of choice for biomedical applications. In addition to XRD and DSC analysis, the barrier properties have been measured through the helium transmission rate and the water vapor transmission rate. Fluorinated parylene films confirmed to be an exceptional material for high temperature applications. The paper is of interest for the broad polymer community, it is well presented, and goes certainly beyond the state of the art in the field.  I recommend publication upon the authors addressing the following minor points:

A)      The chemical strctures are not clear, the authors should redraw them;

B)      Please cite relevant references on fluorinated materials: DOI: 10.1002/adom.202100182; DOI: 10.1021/acs.orglett.0c01043; doi: 10.1002/pola.28532; DOI10.1039/b611336b; doi.org/10.1016/j.progpolymsci.2015.04.007; DOI10.1038/nmat1500

Author Response

Thank you very much for the comments. 

Response to Reviewer 1 Comments

The paper deals with the investigates the thermal impact on bulk properties for four types of parylene films. Parylene, as an organic thin film, is a well-established polymer material exhibiting excellent barrier properties and being often the material of choice for biomedical applications. In addition to XRD and DSC analysis, the barrier properties have been measured through the helium transmission rate and the water vapor transmission rate. Fluorinated parylene films confirmed to be an exceptional material for high temperature applications. The paper is of interest for the broad polymer community, it is well presented, and goes certainly beyond the state of the art in the field.  I recommend publication upon the authors addressing the following minor points:

  1. The chemical structures are not clear, the authors should redraw them;

Response A: We redrew the chemical structure of parylene types. Please find the updated figure in the manuscript.

  1. B)Please cite relevant references on fluorinated materials: DOI: 10.1002/adom.202100182; DOI: 10.1021/acs.orglett.0c01043; doi: 10.1002/pola.28532; DOI10.1039/b611336b; doi.org/10.1016/j.progpolymsci.2015.04.007; DOI10.1038/nmat1500

Response B: Thank you for the aditiional referecens. We intergrated the following relevant references to the bibliography and cites them in the text:

  • org/10.1002/pola.28532.
  • org/10.1002/adom.202100182.
  • doi:10.1039/B611336B

However, the others suggested refrences have been not intergrated because of low revelance with the subject of the manuscript.

Reviewer 2 Report

The research design and the methods used in this study are scientifically sound and showcase the originality of this work, however, I would like to recommend two revisions to improve the flow of the manuscript and connect the conclusions to the results. 

1) Thermal property results could be presented prior to the XRD data. With that order, it would be easier to explain the sudden change in d-spacing for Parylene N samples as well as missing of data points after 250 C for Parylene C. 

2) I think the permeation data collected for oxygen containing environment needs to be separated out as a separate figure and explained better to further support the conclusion. Line 305 'they also exhibit the highest increase...' needs to be rephrased. Based on Fig 2(b), Parylene C film shows the highest increase in domain size and not AF4.

Two minor changes:

1) Please change T (K) in line 168 to T0 (K) to match the equation.

2) Please indicate 'exo up' in the DSC curves. 

Author Response

Response to Reviewer 2 Comments

The research design and the methods used in this study are scientifically sound and showcase the originality of this work, however, I would like to recommend two revisions to improve the flow of the manuscript and connect the conclusions to the results.

1) Thermal property results could be presented prior to the XRD data. With that order, it would be easier to explain the sudden change in d-spacing for Parylene N samples as well as missing of data points after 250 C for Parylene C.

Response 1: We modified the order of the results. Now, thermal properties are presented before the XRD results.

2) I think the permeation data collected for oxygen containing environment needs to be separated out as a separate figure and explained better to further support the conclusion. Line 305 'they also exhibit the highest increase...' needs to be rephrased. Based on Fig 2(b), Parylene C film shows the highest increase in domain size and not AF4.

Response 2: In the first versions of the manuscript draft, the permeation results of oxygen containing environment were separated from the results of nitrogen environment. However, the authors and the internal reviewers suggested presenting all in the same figure in order to be able to compare both environments and to be more concise. Further explanations have been added for a better understanding.

Two minor changes:

  1. A) Please change T (K) in line 168 to T0 (K) to match the equation.

Response A: The modification has been integrated in the updated version of the manuscript.

  1. B) Please indicate 'exo up' in the DSC curves.

Response B: The modification has been integrated in the updated version of the manuscript.

Reviewer 3 Report

After review the manuscript, it is an interesting work, however there are several questions that need to be clarified/corrected before continue the publication process, following specific comments are detailed:

-Define in experimental section the 4 kind of materials that were studied, due that definition was described at the end of Introduction section.

-For DSC why did decide not to carry out a second heating cycle for all materials? this is discussed after in other comment. Also, please define in experimental section which environment was used for this analysis: N2 or Air or O2?

-For TGA analysis, why decide to use a heating rat of 25ºC/min? I mean, this is a rapid heating rate, the common heating rate used is 10ºC/min.

-Figure 2 caption is quite confusing, due indicate that report structural analysis performed at room temperature but variables ( d-spacing and size crystalline domains, are plotted vs temperature.

-In line 216 please clarify which parylene F are referred, due according with codes only one parylene was studied, or if try to refer to a fluorinated compound  please indicate.

-Please explain how the crystalline domain favors the desired application? I men need to explain if crystallinity improve the performance for film application.

-Data reported in table 1 can be observed in figure 2b, so for avoid duplication of data, I recommend to delete table.

-Why in figure 3 parole AF4 report 2nd heating curve? what about other 3 materials? I recommend to report the 2nd heating process for all parylene materials, eve authors indicate that did not reveal additional information, the relevance to carry out a 2nd heating process is to delete the thermal history of materials, and compare if there are significative changes between first and second heating process. Also please indicate the direction for end or exo process in heat flwo curves. Also because for heating curves those process are opposite to melting process which must be endothermic. I mean in lines 242-244 for transitions described at 228 and 281ºC, please indicate if those were described in heating or cooling curve, due if heating curve was used, the description must be changed to melting process of alpha and beta hexagonal, if cooling curve was used, this is correct.

-Please follow a sequence of discussion of figures in order, I mean first describe results in figure 3c and after figure 3b. Also, how can differentiate about degradation of parylene and melting? if melting starts before degradation, is it correct to asseverate that melting of initial structure is carrying out? Furthermore, according with TGA curves, there is not degradation (loss of weigh) of parylene until 400ºC, so this description is incorrect. 

-In figure 5, the temperatures placed top of the bars are confusing,  I recommend to delete them and indicate in definition of color bars to avoid confusion to reader. Furthermore, I recommend to delete this figure due all data reported in it are reported also in table 2, trying to avoid duplication of data.

IN general, the manuscript need to be improved in thermal analysis discussion above all, the work title indicate that Thermal analysis of parylene films is carried out.

Author Response

Dear Reviewer, 

Thank you very much for your comments / corrections. 

Please see the attachment for the replies. 

Best regards

Sébastien Buchwalder

Round 2

Reviewer 3 Report

Thanks to authors for corrections done in the aim to improve the manuscript. Still there are some comments that need to be clarified;

-In comment 8, "are the transition at 228ºC and 281ºC really melting process? Answer is YES, if transition is identified as Endothermic process, if a exothermic process is present, this can not be a melting. About the second heating process, yes I recommend to keep it in thermograms.

-In comment 7 indicate that first version of manuscript did not present table 1 , and internal reviewers suggest to include, but are those reviewers part of review process of journal? I recommend to avoid duplication of data due this is not adequate, which is the justification that include a table and figure with same data?

-About comment 10, again indicate that data reported in table 2 and figure 5 were recommended for internal reviewers, but the publication process is carry out by Polymers Journal. About the proposal for figures, option 2 is good.

Author Response

Dear reviewer, 

Thank you very much for your inputs. 

Please see the attachment for the responses. 

Best regards, 
Sébastien Buchwalder
